# Lymphoid Organ Proteomes Identify Therapeutic Efficacy Biomarkers Following the Intracavitary Administration of Curcumin in a Highly Invasive Rat Model of Peritoneal Mesothelioma

**DOI:** 10.3390/ijms22168566

**Published:** 2021-08-09

**Authors:** Daniel L. Pouliquen, Alice Boissard, Cécile Henry, Stéphanie Blandin, Olivier Coqueret, Catherine Guette

**Affiliations:** 1Université d’Angers, Inserm, CRCINA, F-49000 Angers, France; olivier.coqueret@univ-angers.fr; 2Université d’Angers, ICO Cancer Center, Inserm, CRCINA, F-49000 Angers, France; alice.boissard@ico.unicancer.fr (A.B.); Cecile.Henry@ico.unicancer.fr (C.H.); catherine.guette@ico.unicancer.fr (C.G.); 3Université de Nantes, Plate-Forme MicroPICell, SFR François Bonamy, F-44000 Nantes, France; stephanie.blandin@univ-nantes.fr

**Keywords:** curcumin, proteomics, biomarkers, malignant mesothelioma, rat, spleen, lymph node, immune response, systemic effect

## Abstract

This study aimed to identify the proteomic changes produced by curcumin treatment following stimulation of the host immune system in a rat model of malignant mesothelioma. We analyzed the proteomes of secondary lymphoid organs from four normal rats, four untreated tumor-bearing rats, and four tumor-bearing rats receiving repeated intraperitoneal administrations of curcumin. Cross-comparing proteome analyses of histological sections of the spleen from the three groups first identified a list of eighty-three biomarkers of interest, thirteen of which corresponded to proteins already reported in the literature and involved in the anticancer therapeutic effects of curcumin. In a second step, comparing these data with proteomic analyses of histological sections of mesenteric lymph nodes revealed eight common biomarkers showing a similar pattern of changes in both lymphoid organs. Additional findings included a partial reduction of the increase in spleen-circulating biomarkers, a decrease in C-reactive protein and complement C3 in the spleen and lymph nodes, and an increase in lymph node purine nucleoside phosphorylase previously associated with liver immunodeficiency. Our results suggest some protein abundance changes could be related to the systemic, distant non-target antitumor effects produced by this phytochemical.

## 1. Introduction

Curcumin belongs to a category of bioactive compounds with the propensity to target multiple signaling pathways that are crucial for tumor development [1]. This molecule provides a diversity of interactions explained by its binding to numerous proteins via a unique symmetrical structure combining a central keto-enol tautomer, flexible a,b-unsaturated b-diketone linkers, and terminal *o*-methoxyphenolic groups [2]. In line with the concept of polypharmacology, its mechanism of action suggests that it modulates multiple sensitive nodes belonging to a network of interacting targets [3]. This new concept now offers a strategy to overcome drug resistance [4]. In the case of colon cancer, curcumin represents an epigenetic modulator, cancer stem cell suppressor, and potent autophagy modulator. Trials have demonstrated clinical evidence in prevention and treatment, including synergistic effects on the efficacy of current cancer drugs [5].

Curcumin’s multifaceted effects have already contributed to addressing the chemoresistance and radioresistance commonly found in aggressive cancers such as glioblastoma (GBM) [6]. To circumvent its poor oral absorption following the inclusion of curcumin in the diet, other strategies have been proposed to treat experimental models of GBM or malignant mesothelioma (MM), e.g., intraperitoneal injection [6,7]. Based on proteomic analyses of the liver from curcumin-treated tumor-bearing rats vs. untreated tumor-bearing rats and normal rats, we previously identified quantitative changes in a set of biomarkers representing both determinants of liver colonization and therapeutic targets [8].

It has been reported that an important factor explaining curcumin’s polypharmacology could be the diversity of its degradation products [9]. In our previous studies, successive intraperitoneal administrations of curcumin produced large areas of necrotic/apoptotic tumor cells while inducing an influx of activated CD8+ T cells at the periphery of residual tumors [7] and in the liver [8]. This immune response suggests some common features with the recently described “abscopal effect” (meaning an out-of-field systemic antitumor effect) observed in melanoma patients receiving radiation therapy [10]. Among lymphoid organs, emphasis was given a few years ago to the spleen’s crucial role in regulating immune responses both locally and in the whole body [11]. Therefore, to give insight into the possible biomarkers associated with this effect in curcumin-treated rats, we first investigated the quantitative changes observed in the spleen proteome compared with normal and untreated tumor-bearing rats in our experimental MM model. Cross-comparing spleen proteomic analyses identified eighty-three biomarkers of interest, thirteen of which corresponded to proteins already reported in the literature and involved in curcumin’s therapeutic effects against different cancers. In a second step, cross-comparing spleen data with mesenteric lymph node analyses revealed eight previously undescribed common biomarkers affected by curcumin treatment, which may be related to an abscopal effect.

## 2. Results

### 2.1. Histological Features of the Immune Response in the Spleen and in Residual Tumors of Curcumin-Treated Rats

Three groups of four male rats at four months of age (to mimic the physiological conditions of MM carcinogenesis in adult humans) were used for the study, the experimental design being summarized in Figure 1A. The first group (G1) consisted of normal rats. The second (G2) and third groups (G3) corresponded to rats given a single injection of 3 × 10^6^ M5-T1 tumor cells (tumor challenge) at day 0. G1 and G2 rats received four successive injections of DMSO (50% in saline, i.p., as the vehicle), while G3 rats had four injections of curcumin (1.5 mg/kg in 50% DMSO plus saline) at days 7, 9, 11, and 14. These 8 rats were euthanized at day 16, and the tumors were collected (for treated rats, mostly residual tumors present in the peritoneal cavity). For all twelve rats, spleen and mesenteric lymph nodes were additionally sampled. For both the averaged total tumor mass and spleen mass, a significant decrease was observed in treated (G3) vs. untreated (G2) rats. General histological views of tumor masses (G2) and residual tumors (G3) collected in each rat are shown in Appendix A.

In previous studies, we observed an immune response directed against M5-T1 tumor cells in G3 rats following multiple i.p. curcumin injections, which could be explained by the fact that numerous tumor cells experienced a necrosis/apoptosis process in vivo [7]. Therefore, the decrease in spleen mass observed in G3 vs. G2 rats led us to investigate the differential histological features of the spleen between the three groups (Figure 1B). General views of histological sections of the spleen after HPS staining first revealed a decreased contrast between the white and red pulps of G2 rats, as compared to G1 and G3 rats. In parallel, numerous tumor cells attached to the spleen capsule were specifically present in G2 rats. In a second step, high-magnification views of the white pulp in G3 rats showed the presence of a specific subpopulation of immune cells in the lymphoid sheath surrounding the central artery, which was not observed in G1 and G2 rats. Subsequently, a comparative analysis of the spleen from the G2 and G3 groups after immunostaining with anti-CD8 monoclonal antibody revealed this population corresponded to the CD8+ T cells specifically observed in G3 (Figure 1B, bottom rank, yellow arrows), which were larger and more intensely stained than their G2 counterparts. In a third step, high-magnification views of these areas confirmed that G2 rats were specifically characterized by smaller, more heterogeneous, and irregularly shaped cells, while the lymphoid sheath cells in G3 rats were nicely individualized (Figure 1C, top rank). Examination of red pulps also revealed the same specific feature (Figure 1C, bottom rank, arrow). Finally, residual tumor masses collected in the peritoneal cavity of G3 rats were also characterized by massive infiltration of their surrounding environment and connective tissue separating muscle fibers with activated lymphocytes presenting the same morphology (as previously observed in the liver [8]), a finding never obtained in G2 rats (Figure 1D).

### 2.2. Spleen Biomarkers of Tumor Progression and Curcumin-Induced Immune Response

We detected 1483 proteins in each spleen sample analyzed by SWATH-MS. The number of proteins exhibiting significant abundance differences (*p* < 0.05) in G2 vs. G1, G3 vs. G2, and G3 vs. G1 were 285, 192, and 250, respectively. Cross-comparing the three files led to a list of 83 proteins (Table 1) common to the 285 proteins found in G2 vs. G1 and the 192 proteins found in G3 vs. G2, of which 60 proteins were excluded from the 250 proteins found in G3 vs. G1 (satisfying the condition *p* < 0.05).

Among the first group of proteins, 55 exhibited an increase (including 4 ribosomal proteins) and 5 a decrease linked to tumor progression, which returned to normal values after curcumin treatment (Table 1). A second group of 14 proteins was also common to G2 vs. G1 and G3 vs. G2 (still evolving in opposite directions), while showing significant abundance changes (*p* < 0.05) between G3 and G1, including 12 presenting an increase and 2 a decrease. Finally, a third group of nine proteins presented abundance changes evolving in the same direction in G2 vs. G1 and G3 vs. G2 (with *p* < 0.05 between G3 and G1), including two showing an increase while seven a decrease (Table 1). Figure 2A shows the main subcellular and extracellular locations of the 83 proteins listed in Table 1. A gene ontology analysis of upregulated proteins, conducted with DAVID Bioinformatics Resources 6.8 (https://david.ncifcrf.gov) (accessed on 15 July 2021) revealed two main charts: (1). proteins involved in cell-cell adhesion (with a *p* value of 5.5 × 10^−3^), and (2). Extracellular Matrix (ECM) proteins (with a *p* value of 8.5 × 10^−4^). Downregulated proteins did not provide any information due to their restricted number. Moreover, a functional analysis of known and predicted protein–protein interactions for the 55 upregulated proteins, using the STRING Database (https://string-db.org) (accessed on 15 July 2021), showed 31 were associated in a single network, with 2 others being associated separately while 22 presented no interactions (Figure 2B).

We next analyzed which of the eighty-three proteins listed in Table 1 were already reported as being associated with curcumin’s therapeutic effects in the literature. We found thirteen proteins reported in PubMed, three of which were cited more than four times over a 25-year period and encoded by *Ahr*, *Pdcd4*, and *Rhoa* (Table 2). The quantitative changes we observed in the spleen for these three proteins are illustrated in Figure 3A. For PDCD4, immunohistochemical staining of the spleen also revealed the presence of clusters of densely stained PDCD4+ cells in the lymphoid sheaths surrounding the central arteries of the splenic nodules in G2 rats, which were rarely found in G3 rats (Figure 3B,C).

### 2.3. Common and Specific Biomarkers of Curcumin-Induced Effects in Spleen and Lymph Nodes

We next determined which of the eighty-three biomarkers identified in Table 1 exhibited similar profiles in another lymphoid organ located at distance from the initial place of M5-T1 tumor development, the omentum, which has an anatomical relationship with the spleen. Therefore, we investigated the proteome of mesenteric lymph nodes, as these are numerous large masses allowing proteomic analyses. Representative examples of the histological features of lymph nodes from each group of rats are given in Appendix A. We detected 1498 proteins in each lymph node sample analyzed by SWATH-MS. The number of proteins exhibiting significant abundance changes (*p* < 0.05) in G2 vs. G1, G3 vs. G2, and G3 vs. G1 were 415, 256, and 188, respectively.

In a first step, cross-comparing the proteins not common to these lists and Table 1 led us to define a repertoire of specific spleen biomarkers. The proteins exhibiting the largest abundance changes in this organ are illustrated in Figure 4.

In a second step, eight proteins showing similar profiles of abundance changes in spleen and mesenteric lymph nodes (common to the lists G2 vs. G1 and G3 vs. G2 above and to Table 1) were identified. Seven of these proteins were excluded from the 188 proteins found in G3 vs. G1 (satisfying the condition *p* > 0.05), encoded by *Fubp1*, *Hnrnpa2b1*, *Hsd17b10*, *Mybbp1a*, *Psmb10*, *Purb*, and *Wipf1*. The protein encoded by *Impa1* was the only one satisfying the condition *p* < 0.05 for the comparison G3 vs. G1. Two additional proteins showed tendencies for the same, encoded by *Cpne1* and *Mri1* genes (Appendix A). Figure 5 shows the comparison of abundance changes for both spleen and lymph nodes for these eight proteins.

In a third step, the analysis of specific biomarkers of curcumin-induced effects in lymph nodes identified a single protein, purine nucleoside phosphorylase, encoded by the *Pnp* gene (Figure 6A). This protein exhibited the same pattern of abundance changes as previously observed in the liver of the same rats [8]. Moreover, immunohistochemical staining of lymph nodes revealed the presence of higher stained PNPH+ cells in the medullary sinus in G3 vs. G2 rats (Figure 6B), and of PNPH+ cells in the two parts of the cortex of lymphatic nodules in G3 rats, which were absent in G2 rats (Figure 6C).

### 2.4. Tumor Progression and Curcumin-Induced Effects on Plasma Biomarkers

Among the second group of biomarkers listed in Table 1, in addition to complement C9, for which the increase observed in the spleen and induced by tumor progression was completely reversed following curcumin treatment (Figure 4), seven other plasma proteins exhibited a rise in abundance that was partially reversed in G3 rats (Figure 7A). These included five positive acute-phase proteins (encoded by *Cp*, *C3*, *Crp*, *Fga*, and *Fgb* genes) for which elevated plasma levels are generally associated with inflammation. The two others were hemopexin (encoded by *Hpx*) involved in the scavenging of heme, and fibronectin (encoded by *Fn1*), a component of the extracellular matrix playing a major role in cell adhesion and migration. In lymph nodes, the only protein showing an evolution comparable to that observed in the spleen was the C-reactive protein (Figure 7B), a decreased level in complement C3 also being induced after curcumin treatment.

## 3. Discussion

Curcumin’s effects on all stages of tumorigenesis are well established. However, although its action on multiple cell-signaling pathways has been the subject of numerous studies on cancer cells in vitro, several questions remain on its complex mechanisms of action in vivo. To shed light on this point, we used a rat model of aggressive peritoneal mesothelioma [7] and an established treatment procedure with curcumin injected intraperitoneally to determine the secondary lymphoid organs’ implication in the immune response induced, and to further identify potential associated biomarkers [8]. Taverna and colleagues previously reported that innovative quantitative proteomic techniques such as SWATH-MS helped reveal how curcumin modulated the composition of exosomes released by cancer cells, making it possible to reverse their pro-angiogenic activity [42]. As reviewed by Bronte and Pittet, investigations on mouse spleen functions have unveiled a wider role than previously expected, especially in systemic regulation of immunity [11]. In line with its crucial implication in immune responses, which is deleterious to the host in metastatic cancers, this organ was shown to represent the site of accumulation of immature myeloid-derived suppressor cells (MDSCs) in tumor-bearing animals [11]. Tu and colleagues have revealed that one of curcumin’s most impressive effects in vivo was the inhibition of these myeloid cells’ expansion in the spleen and polarization toward a M1-like phenotype [43]. Therefore, our work first aimed at providing information at a molecular level on spleen proteomic biomarkers which were affected by both tumor progression and curcumin treatment. To date, this field of research has been poorly documented experimentally, and limited to the case of non-medicated and non-immunized rats [44].

Among the 55 proteins showing significant abundance changes common to G2 vs. G1 and G3 vs. G2, and excluded from G3 vs. G1, 13 were already reported in the literature as being associated with curcumin’s effects. One especially has been the subject of numerous studies since 1998 [12], related to curcumin’s suppression of the transformation of the aryl hydrocarbon receptor (encoded by the *Ahr* gene) in cancer cells [14,20]. A recent study by Mohammadi-Barbori and colleagues has shown that this effect’s mechanistic aspects also include chromatin remodeling [19] in addition to a previously documented modification of the cellular redox status [15]. Interestingly, another important target is represented by miRNA expression [30], and particularly miRNA-21 (miR-21), which is regulated by curcumin [27,29,31,32]. Even more interesting, *Pdcd4* represents a functional target of miR-21 [45], while the anticarcinogenic effects of curcumin were reported to be partly mediated through modulation of miR-21 expression [32]. These two findings suggest that the decrease in PDCD4 immunostaining we observed in spleen cells of curcumin-treated rats vs. untreated rats might reflect the reversal produced by curcumin treatment on the deleterious effects induced by the tumor development on immune cell functions. RhoA, one of the most extensively studied members of the Rho family of small GTPase, is also involved in tumor cell migration and invasion [46], as well as fibrosis induction [36,37], both of which are prevented by curcumin through reversion of the epithelial–mesenchymal transition [35,38]. Finally, in our study, among the twelve proteins involved in the most significant abundance changes specifically observed in the spleen, which were related to tumor progression and completely reversed upon curcumin treatment, two already reported in the literature concerned complement C9 and transgelin-2. In the first case, Jacob and colleagues have previously reported the reduction of complement activation and C9 deposits by curcumin [22]. In our data, the fact that no change was observed in parallel in lymph nodes could be explained by the finding that activation of the complement cascade resulted in rapid disappearance of C9 from the plasma and accumulation in the spleen [47]. However, although the reversion induced by curcumin treatment was not complete, a common effect in spleen and lymph nodes was observed for another member of the complement system, C3, consistent with its known role in both the classical and alternative pathways [48]. Regarding transgelin-2, Ma and colleagues have revealed that the gene encoding for this protein belonged to a group of fifteen genes involved in curcumin’s therapeutic effect against human tongue cancer [41].

The decrease in carbonic anhydrase 1 (encoded by *Ca1*) and delta-aminolevulinic acid dehydratase (encoded by *Alad*) induced in the spleen by tumor progression concurred with our histological observations on the red pulp, as these enzymes are specific to erythroid cells. However, in addition to their balance, both components of the spleen appeared to be affected in G2 rats. The parallel decrease in the activity of nucleosome assembly protein 1-like 1 (encoded by *Nap1l1*) specific to this group could be explained by Tanaka and colleagues’ findings that its downregulation renders the cell vulnerable to apoptotic cell death via attenuation of NF-kB transcriptional activity on *Mcl-1* [49]. Indeed, the protein is quite abundant in immune cells. Moreover, this observation agrees with another report from Çevik et al., demonstrating that lack of NAP1L1 leads to inefficient TLR3 responses, which are involved in pathogen recognition and activation of innate immunity [50]. Among the nine other main spleen proteins concerned with a complete reversion of the increase produced by tumor progression, we found two proteins involved in cell transduction pathways, one located in the cytosol (encoded by *Prkar1a*), and the second in the nucleus (encoded by *Mapk3*). The interrelationship between cyclic AMP elevation and ERK1/2 activation has previously been documented in cancer cells [51], suggesting that a combined increase in untreated tumor-bearing rats could be associated with early tumor progression in this lymphoid organ. Another category of proteins exhibiting increased abundance in G2 rats includes calreticulin and leucine-rich repeat-containing protein 59, both located in the endoplasmic reticulum (ER). This observation could relate to the fact that enhanced ER activity is required to facilitate the folding, assembly, and transportation of membrane and secretory proteins, all these functions being carried out by chaperones [52]. However, calreticulin can also be co-located in the nucleus, elevated levels being associated with additional cellular processes, and particularly with poor outcomes in some cancers as reviewed by Fucikova et al., in line with enhanced angiogenesis and facilitation of the migration and proliferation of tumor cells [53]. As in our study, elevated levels were observed not in the tumor, but a secondary lymphoid organ, the spleen, perhaps due to efferocytosis, which corresponds to the uptake and removal of apoptotic cells by phagocytes leading to silenced immune responses [54]. The elevated levels of annexins A1 and A5 are also frequently associated with tumor invasiveness [55,56], while the ephrin receptor family plays a major role in modifications of the tumor microenvironment and tumor immune evasion [57]. In this context, the increase we observed in the peculiar case of ephrin-B1 in the spleen could be explained by Iwasaki et al.’s finding that activation of this protein’s expression is related to chronic hypoxia [58], which is also consistent with our observations on the aforementioned decrease in erythrocytic cell biomarkers.

In our study, cross-comparing spleen and lymph node proteomic data led us to identify several common biomarkers of interest, among which C-reactive protein, which represents one member of the positive acute-phase response proteins commonly elevated in inflammatory diseases and cancers. Even more interestingly, eight biomarkers showed similar profiles of abundance changes in both secondary lymphoid organs. The protein exhibiting the most significant changes, encoded by *Mybbp1a*, was initially reported to belong to Aurora kinases, presenting an essential role in the normal progression of mitosis [59]. Subsequently, George and colleagues pointed to the critical role of MYBBP1A in the regulation of senescence in cancer cells under genotoxic stress [60], while Weng et al. emphasized its overexpression in the progression of hepatocellular carcinoma with poor prognosis [61]. Interestingly, Nahálkova recently reported that MYBBP1A is associated with p53, TPPII, SIRT6, and CD47 in a protein interaction network that controls the Warburg effect [62]. Two other proteins of interest that bind to single-stranded DNA in the nucleus are encoded by *Purb* and *Fubp1*. In the first case, purine-rich element-binding proteins A and B are implicated in the regulation of gene expression at both transcription and translation levels, the highest levels of PURB being observed in myeloid cells from patients with primary acute myelogenous leukemia displaying risk factors forecasting a poor prognosis [63]. At the end of the 2010s, FUBP1 represented one of the fifty types of specific factors already reported to regulate *c-myc* transcription [64], whose overexpression was associated with the regulation of proliferation and migration in liver cancer cells [65]. By 2019, Debaize and Troadec concluded that FUBP1 represented a potent pro-proliferative and anti-apoptotic factor which also appeared to be of clinical relevance in oncogenesis [66]. Recent findings by Ma and colleagues have confirmed this protein’s value as a novel prognosis factor and therapeutic target for cervical carcinoma [67]. Another protein showing extremely significant abundance changes in both organs is WIPF1, which belongs to the WASP-interacting family of proteins [68] and is involved in the formation of actin-rich membrane protrusions degrading the extracellular matrix called invadopodia [69]. Aberrant expression of WIPF1 has been reported in several cancers, contributing to invasive and metastatic properties [70]. In the context of our study, an intriguing feature is the observation made by Ramesh et al. concerning the implication of WIPF1 in the regulation of TCR signaling, linked to cytoskeletal abnormalities in inherited immune deficiency [71]. Finally, two more important members are ROA2 and PSB10. The former is an RNA-binding protein of extracellular vesicles involved in intercellular communication [72,73], while the latter represents a key subunit of the immunoproteasome [74,75].

Lastly, the pattern of abundance changes that we observed specifically in lymph nodes for purine nucleoside phosphorylase, which mimicked our previous finding in the liver of the same rats [8], tends to confirm its value as an immune deficiency biomarker.

## 4. Materials and Methods

### 4.1. Experimental Procedures in Rats

F344 male Fisher rats at seven weeks of age were purchased from Charles River Laboratories (L’Arbresle, 69, France) and maintained under SPF (specific-pathogen-free) status in the UTE-IRS UN Animal Facility of the University of Nantes following European Union guidelines for the care and use of laboratory rodents in research protocols. The experiments were approved by the Ethics Committee for Animal Experiments (CEEA) of the Pays de la Loire Region of France (2011.38) and #01257.03 (approved on 19 June 2015) of the French Ministry of Higher Education and Research (MESR). For groups G2 and G3, the M5-T1 original neoplastic cell line [7] was injected intraperitoneally (3 × 10^6^ cells in 200 µL PBS buffer) at day 0 (Figure 1A). At day 16, all rats were anesthetized in an isoflurane chamber (Forene^®^, Abbott, Rungis cedex, France) and finally euthanized in their home cage with a rate of 30% volume displacement per minute of CO_2_. In untreated rats (G2), numerous metastatic nodules were collected in the peritoneal cavity, invading the liver, pancreas, diaphragm, gut, and parietal peritoneum, easily identified by their white color and dense texture. In curcumin-treated rats (G3), metastatic nodules were absent from the peritoneal cavity and diaphragm, and a few residual tumor masses were collected in the peritoneal cavity, attached to the omentum, liver, or gut (Appendix A).

### 4.2. Histological Analyses

Tissue samples (tumor, spleen, and mesenteric lymph nodes) for histological analyses were collected from the three groups of rats (Figure 1A), fixed in 4% paraformaldehyde (in PBS buffer), embedded in paraffin, and then cut with a Leica RM2255 microtome (Leica Biosystems, Nussloch, Germany). For histological examination, 3-µm-thick sections of all samples were stained with hematoxylin-phloxine-saffron (HPS), and the slides were scanned (NanoZoomer 2.0 HT Hamamatsu, Japan). For T cell characterization, spleen sections were stained with anti-CD8 (LS-B3665, LSBio France, Nanterre, France) monoclonal antibody. Complementary immunohistochemical analyses were conducted with rabbit anti-PDCD4 (NBP1-76738) and anti-PNPH (NBP1-82541) antibodies (Novus Biologicals, Centennial, CO, USA).

### 4.3. Proteomic Analyses

For SWATH-MS analyses, four 20 µm-thick sections of each spleen and lymph node sample were used. In a first step, HPS-stained sections were examined to select areas of interest (to remove any risk of tumor cell contamination), and then corresponding areas were removed with a scalpel and collected in a 1.5-mL Eppendorf^®^ microtube. The samples were deparaffinized with xylene washes, rehydrated in graded ethanol solutions, and vacuum-dried [8]. Dried tissues were resuspended in 200 µL of Rapigest SF (Waters, Milford, MA, USA) and dithiothreitol added to a final concentration of 5 mM (DTT, AppliChem, Darmstadt, Germany). Tubes were incubated in a thermo shaker at 95 °C for one hour, and sonication was performed twice by Ultrasonic processor 75185 (130 W, 20 KHz, Bioblock Scientific, Illkirch, France). Subsequently, cysteine residues were alkylated by adding 200 mM S-methyl-methanethiosulfonate (MMTS) to a final concentration of 10 mM (incubated at 37 °C). Sequencing-grade trypsin was added in a ratio ≥ 2 µg mm^−3^ tissue (incubated at 37 °C overnight). The reaction was stopped with formic acid (9% final concentration) and incubated at 37 °C for one hour, and the acid-treated samples were centrifuged at 16,000× *g* for ten minutes. Salts were removed from the supernatant and collected into new reaction microtubes using self-packed C18 STAGE tips. Peptide concentrations were finally determined with the Micro BCA™ Protein Assay Kit (Thermo Fisher Scientific, Saint-Herblain, 44, France).

Five micrograms of each sample were analyzed by repeating the whole gradient cycle corresponding to the acquisition of 32 time-of-flight MS/MS scans of overlapping sequential precursor isolation windows (25 *m*/*z* isolation width, 1 *m*/*z* overlap, high-sensitivity mode). Each MS/MS scan covered the 400 to 1200 *m*/*z* mass range, with a previous MS scan for each cycle. The accumulation time was 50 ms for each MS scan and 100 ms for the product ion scan (230 to 1500 *m*/*z*), leading to a total cycle time of 3.5 s. Peak extraction of the SWATH data and relative quantification were further performed as previously described [8].

## 5. Conclusions

Taken together, our observations suggest that multiple intraperitoneal administration of curcumin produced effects in secondary lymphoid organs of tumor-bearing rats at two complementary levels. At the spleen level, close to the initial location of tumor development, curcumin restored both red and white pulp functions and the delicate balance observed between these two compartments in normal animals, shown by the return to normal abundances of 53 proteomic biomarkers. In lymph nodes, the decrease in PNPH produced by tumor development and related to immune deficiency also returned to normal under curcumin treatment. Furthermore, curcumin contributed to reducing the elevation of twelve additional biomarkers, and to limiting the reduction of two others produced by tumor progression in untreated animals. At a second level, the fact that eight biomarkers showed similar profiles of abundance changes in the spleen and mesenteric lymph nodes, far from the place of tumor development, tends to demonstrate the existence of an out-of-field systemic antitumor effect, probably related to massive induction of necrosis/apoptosis in tumor cells by curcumin treatment. The nature of biomarkers affected by this “abscopal effect” finally suggests, in addition to C-reactive protein secreted in the blood and related to inflammation, an action at multiple levels in immune cells within lymphoid follicles, including transcription regulation, reorganization of the cytoskeleton, TCR signaling, and regulation of proteasomal activity. These observations provide a good basis for future mechanistic studies.

## Figures and Tables

**Figure 1 ijms-22-08566-f001:**
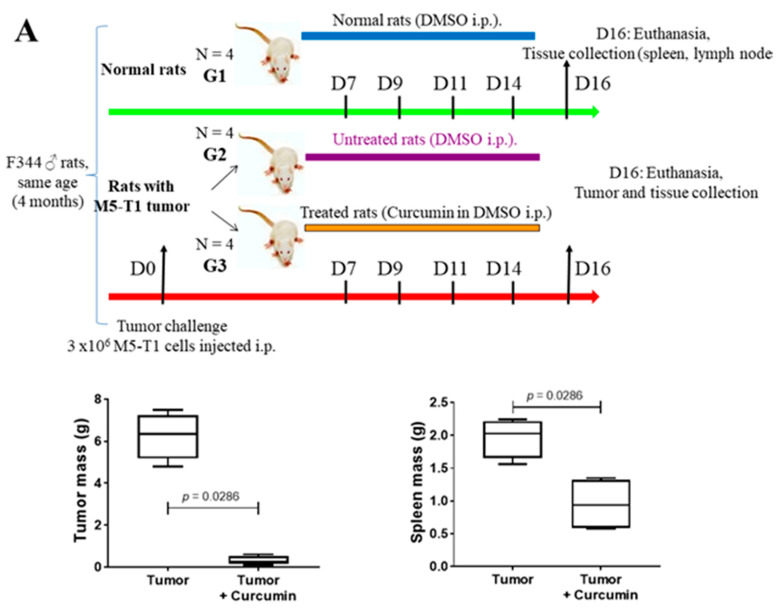
Spleen histological changes induced by M5-T1 tumor progression or curcumin treatment. (**A**) Experimental procedure showing how tumor and lymphoid organs were sampled for proteomic and histological analysis (top), and the effect of curcumin treatment on tumor and spleen mean masses. (**B**) Identification of a subpopulation of activated lymphocytes showing specific morphological features in curcumin-treated rats (G3 group). Top rank, general views of the spleen from one representative rat of each group (HPS staining, ×15, the scale bar represents 1 mm). Middle rank, anti CD8+ IHC staining, comparison of G3 (right) vs. G2 rat (left), ×50, the scale bar represents 500 µm. Bottom rank, magnification of a lymphoid nodule, anti CD8+ IHC staining, showing a significant concentration of CD8+ T cells into the lymphoid sheath surrounding the central artery (right), and corresponding HPS staining (left) revealing the presence of bigger, densely stained lymphocytes in the corresponding area (yellow arrows), ×200, the scale bar represents 100 µm. (**C**) High-magnification views (HPS staining) of lymphoid sheaths (top rank) and red pulps (bottom rank, the yellow arrow points to a small cluster of large lymphocytes), the scale bar represents 25 µm. (**D**) Comparison of histological sections (HPS staining) of M5-T1 tumors, G3 (curcumin-treated rat, top rank) vs. G2 group (untreated rat, bottom rank). Top, general view (left, the scale bar represents 1 mm), and magnifications of two areas illustrated by the yellow rectangles showing clusters of immune cells infiltrating the connective tissue separating muscle fibers (insert, the scale bar represents 50 µm) and the external part of the tumor (right, the scale bar represents 25 µm). Bottom, general view (the scale bar represents 1 mm), and magnification of a large area (the scale bar represents 100 µm) showing numerous tumor cells invading the muscle free of any lymphocyte cluster.

**Figure 2 ijms-22-08566-f002:**
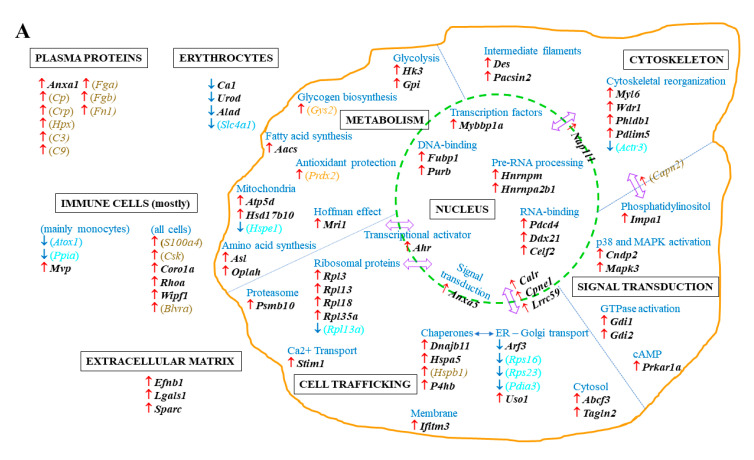
(**A**) Main locations of the eighty-three potential targets of curcumin’s therapeutic efficacy in the spleen. The proteins illustrated in this figure correspond to the list detailed in Table 1. Each protein is identified by the gene (*in italics*) encoding its expression. Names in black and bold represent the first group of proteins for which a *p* value < 0.05 was observed in both G2 vs. G1 and G3 vs. G2, with abundance changes evolving in opposite directions, respectively (arrows indicating changes observed in G2 vs. G1, **↑** for increase, **↓** for decrease), while differences were not significant in G3 vs. G1 (*p* value > 0.05). Names between brackets correspond to the second group of proteins satisfying the same conditions in G2 vs. G1 and G3 vs. G2, while showing significant differences in G3 vs. G1 (*p* value < 0.05), written in brown for increase and blue for decrease. Proteins of the third group (changes evolving in the same directions in both G2 vs. G1 and G3 vs. G2) are represented by names in brackets in orange and cyan blue, for increase and decrease, respectively. Locations were recorded on https://www.proteinatlas.org (accessed on 15 July 2021). (**B**) Analysis of protein–protein interactions (PPIs) by online bioinformatics (STRING Database, https://string-db.org) (accessed on 15 July 2021), restricted to the 55 proteins showing abundance increase in G2 vs. G1, decrease in G3 vs. G2, while unchanged in G3 vs. G1.

**Figure 3 ijms-22-08566-f003:**
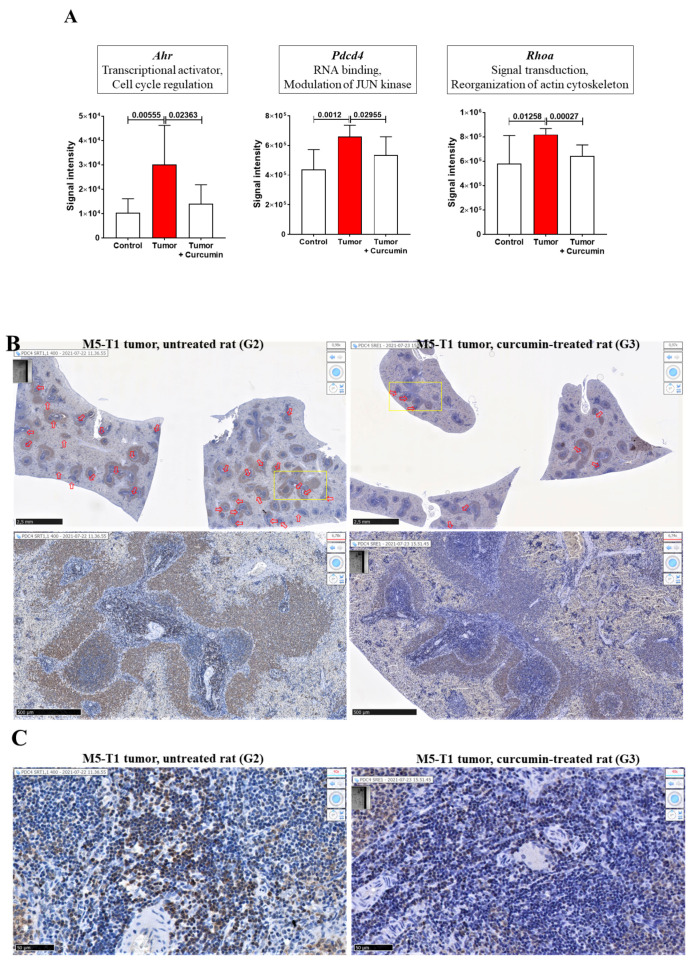
Abundance changes in the spleen. (**A**) Proteomic analysis of three proteins most reported in the literature as being associated with curcumin’s therapeutic effects. Gene names encoding each protein are written in italics at the top of each graph. Left to right: G1 (control = normal rats), G2 (tumor = untreated rats), and G3 (curcumin-treated rats); *p* values of the comparison between groups are indicated at the top of the bars. (**B**) Immunohistochemical staining with anti-PDCD4 antibody of the spleen from two representative rats of G2 (left) and G3 (right) groups. Top, general views, open red arrows show the locations of clusters of positive cells in the lymphoid sheaths surrounding the central arteries of the splenic nodules, the scale bars represent 2.5 mm. Bottom, enlargement of the two areas represented by the yellow rectangles showing differences in the staining intensity of these clusters in G2 vs. G3; the scale bars represent 500 µm. (**C**) High magnification views of these two areas showing the differences in staining intensity, morphology, and cell density in G2 vs. G3; the scale bars represent 50 µm.

**Figure 4 ijms-22-08566-f004:**
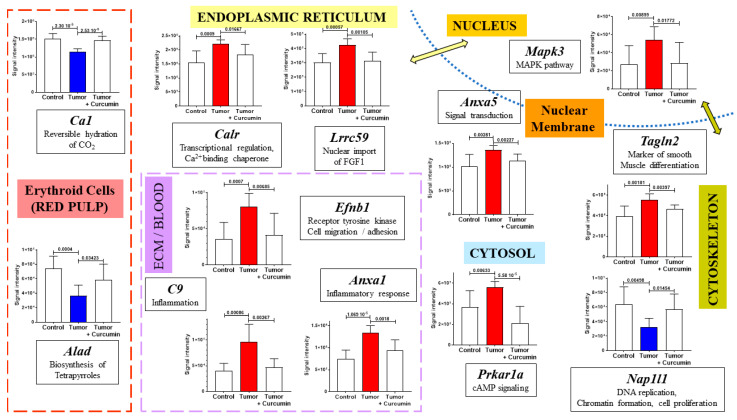
Most significant abundance changes specifically observed in the spleen. Gene names encoding each protein are written in bold and italics for each graph. The green and yellow arrows indicate proteins located in both the nucleus and another subcellular compartment (locations were recorded on https://www.proteinatlas.org) (accessed on 19 July 2021).

**Figure 5 ijms-22-08566-f005:**
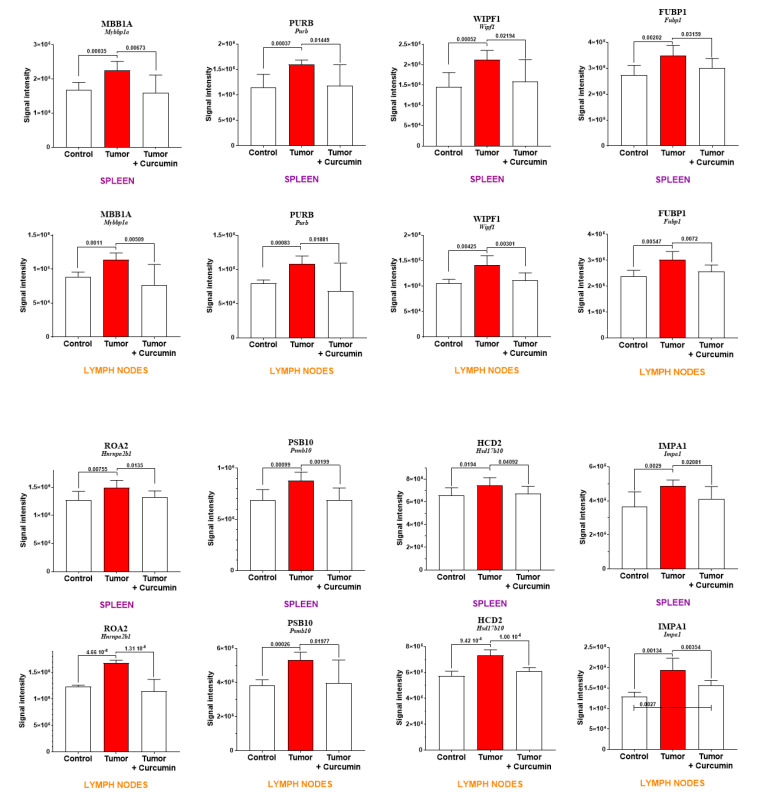
Proteins exhibiting similar profiles of abundance changes in the spleen and lymph nodes. Gene names encoding each protein are in italics and protein codes (for *Rattus norvegicus*) in upper case and bold, as indicated at the top of each graph.

**Figure 6 ijms-22-08566-f006:**
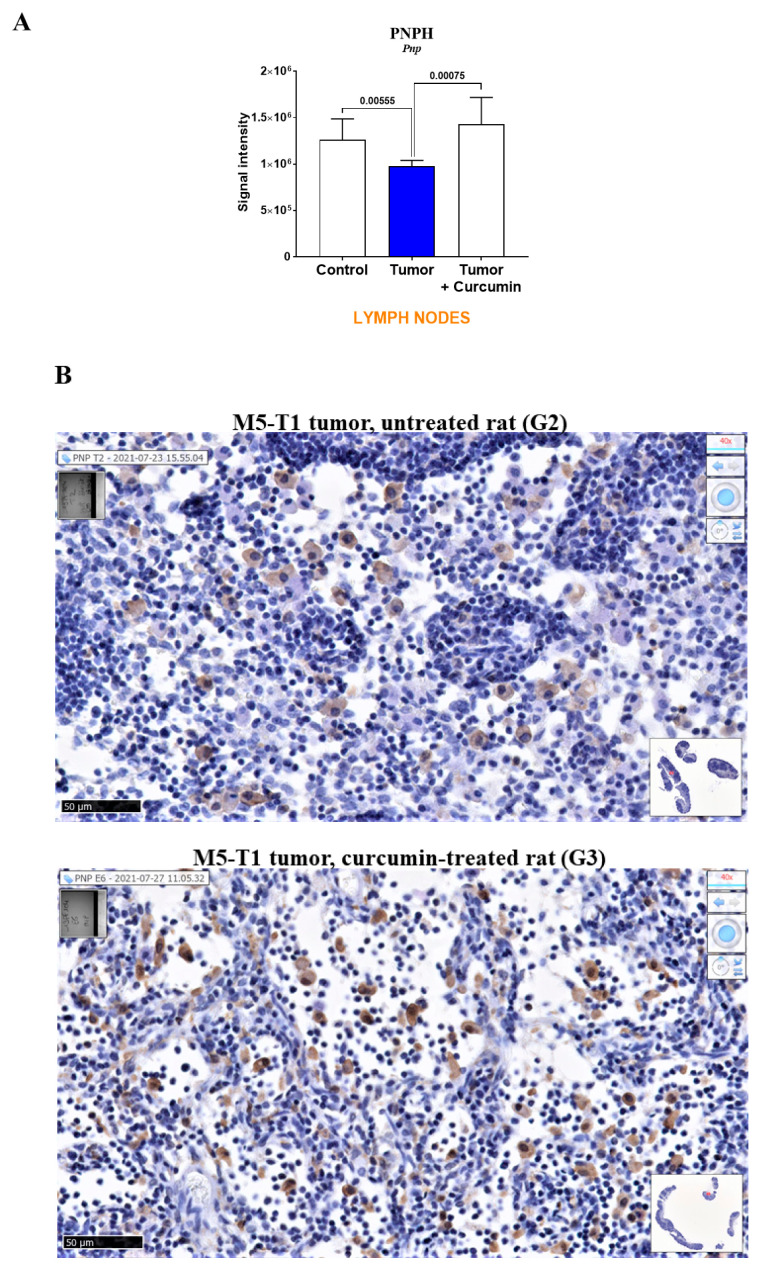
Purine nucleoside phosphorylase (PNPH) in lymph nodes. (**A**) Proteomic analysis. (**B**,**C**) Immunohistochemical staining with anti-PNPH antibody of lymph nodes from two representative rats of G2 (top) and G3 (bottom) groups (general views in inserts). Photographs in (**B**) illustrate the differences in staining intensity of PNPH+ cells present in the medullary sinus. Photographs in (**C**) show the presence of numerous PNPH+ cells in the outer cortex and some in the inner cortex in G3 rat (absent in G2 rat). Yellow arrows indicate probable macrophages/dendritic cells in the germinal center. Scale bars represent 50 µm.

**Figure 7 ijms-22-08566-f007:**
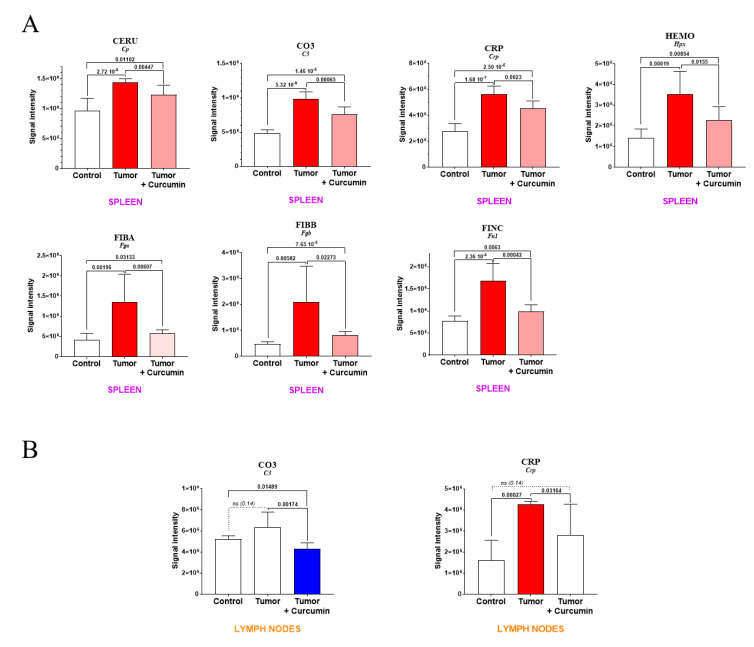
Levels of plasma biomarkers in the spleen (**A**) and lymph nodes (**B**). *p* values of the comparison between groups are indicated at the top of the bars. Red (light red representing the partial decrease between “Tumor + curcumin” = G3 vs. “Tumor” = G2) and blue bars representing the significant decrease relative to (Control = G1), respectively. ns and dotted line = non-significant difference.

**Table 1 ijms-22-08566-t001:** List of spleen proteins exhibiting significant abundance changes (*p* < 0.05) common to the two comparisons: group 2 (G2, untreated tumor-bearing rats) vs. group 1 (G1, normal rats), and group 3 (G3, tumor-bearing rats treated with curcumin) vs. group 2. Non-significant differences for G3 vs. G1 = ns (*p* > 0.05). Gene names (*in italics*) are given for *Rattus norvegicus*. **↑** increase, **↓** decrease.

Code	Gene	Full Name	G2/G1	G3/G2	G3/G1
AACS	*Aacs*	Acetoacetyl-CoA synthetase	**↑** *p* = 0.01991	**↓** *p* = 0.02754	ns (*p* = 0.69)
ABCF3	*Abcf3*	ATP-binding cassette sub-family F member 3	**↑** *p* = 0.0423	**↓** *p* = 0.0127	ns (*p* = 0.21)
AHR	*Ahr*	Aryl hydrocarbon receptor	**↑** *p* = 0.00555	**↓** *p* = 0.02363	ns (*p* = 0.29)
ANXA1	*Anxa1*	Annexin A1	**↑** *p* = 1.69 10^−5^	**↓** *p* = 0.0018	ns (*p* = 0.11)
ANXA5	*Anxa5*	Annexin A5	**↑** *p* = 0.00281	**↓** *p* = 0.00227	ns (*p* = 0.27)
ARLY	*Asl*	Argininosuccinate lyase	**↑** *p* = 0.01787	**↓** *p* = 0.00459	ns (*p* = 0.65)
ATPD	*Atp5d*	ATP synthase subunit delta, mitochondrial	**↑** *p* = 0.03331	**↓** *p* = 0.00153	ns (*p* = 0.53)
CALR	*Calr*	Calreticulin	**↑** *p* = 0.0009	**↓** *p* = 0.01667	ns (*p* = 0.17)
CELF2	*Celf2*	CUGBP Elav-like family member 2	**↑** *p* = 0.04365	**↓** *p* = 0.03967	ns (*p* = 0.25)
CNDP2	*Cndp2*	Cytosolic non-specific dipeptidase	**↑** *p* = 0.01447	**↓** *p* = 0.02765	ns (*p* = 0.16)
CO9	*C9*	Complement component C9	**↑** *p* = 0.00086	**↓** *p* = 0.00267	ns (*p* = 0.45)
COR1A	*Coro1a*	Coronin-1A	**↑** *p* = 0.00729	**↓** *p* = 0.00078	ns (*p* = 0.79)
CPNE1	*Cpne1*	Copine-1	**↑** *p* = 0.02077	**↓** *p* = 0.00269	ns (*p* = 0.35)
DDX21	*Ddx21*	Nucleolar RNA helicase 2	**↑** *p* = 0.00579	**↓** *p* = 0.02476	ns (*p* = 0.22)
DESM	*Des*	Desmin	**↑** *p* = 0.01363	**↓** *p* = 0.01693	ns (*p* = 0.51)
DJB11	*Dnajb11*	DnaJ homolog subfamily B member 11	**↑** *p* = 0.02069	**↓** *p* = 0.00836	ns (*p* = 0.85)
EFNB1	*Efnb1*	Ephrin-B1	**↑** *p* = 0.0007	**↓** *p* = 0.00685	ns (*p* = 0.67)
FUBP1	*Fubp1*	Far upstream element-binding protein 1	**↑** *p* = 0.00202	**↓** *p* = 0.03159	ns (*p* = 0.14)
G6PI	*Gpi*	Glucose-6-phosphate isomerase	**↑** *p* = 0.03101	**↓** *p* = 0.00557	ns (*p* = 0.07)
GDIA	*Gdi1*	Rab GDP dissociation inhibitor alpha	**↑** *p* = 0.00306	**↓** *p* = 0.01494	ns (*p* = 0.23)
GDIB	*Gdi2*	Rab GDP dissociation inhibitor beta	**↑** *p* = 0.01183	**↓** *p* = 0.0371	ns (*p* = 0.34)
GRP78	*Hspa5*	78kDa glucose-regulated protein	**↑** *p* = 0.00475	**↓** *p* = 0.00064	ns (*p* = 0.21)
HCD2	*Hsd17b10*	3-hydroxyacyl-CoA dehydrogenase type-2	**↑** *p* = 0.0194	**↓** *p* = 0.04092	ns (*p* = 0.70)
HNRPM	*Hnrnpm*	Heterogeneous nuclear ribonucleoprotein M	**↑** *p* = 0.00659	**↓** *p* = 0.0007	ns (*p* = 0.54)
HXK3	*Hk3*	Hexokinase-3	**↑** *p* = 0.03094	**↓** *p* = 0.00115	ns (*p* = 0.96)
IFM3	*Ifitm3*	Interferon-induced transmembrane protein 3	**↑** *p* = 0.00314	**↓** *p* = 0.0218	ns (*p* = 0.95)
IMPA1	*Impa1*	Inositol monophosphatase 1	**↑** *p* = 0.0029	**↓** *p* = 0.02081	ns (*p* = 0.29)
KAP0	*Prkar1a*	cAMP-dependent protein kinase type I-alpha regulatory subunit	**↑** *p* = 0.00633	**↓** *p* = 5.58 10^−5^	ns (*p* = 0.08)
LEG1	*Lgals1*	Galectin-1	**↑** *p* = 0.01128	**↓** *p* = 0.00304	ns (*p* = 0.30)
LRC59	*Lrrc59*	Leucine-rich repeat-containing protein 59	**↑** *p* = 0.00057	**↓** *p* = 0.00105	ns (*p* = 0.76)
MBB1A	*Mybbp1a*	Myb-binding protein 1A	**↑** *p* = 0.00035	**↓** *p* = 0.00673	ns (*p* = 0.71)
MK03	*Mapk3*	Mitogen-activated protein kinase 3	**↑** *p* = 0.00899	**↓** *p* = 0.01772	ns (*p* = 0.91)
MTNA	*Mri1*	Methylthioribose-1-phosphate isomerase	**↑** *p* = 0.00444	**↓** *p* = 0.0179	ns (*p* = 0.86)
MVP	*Mvp*	Major vault protein	**↑** *p* = 0.00743	**↓** *p* = 0.02179	ns (*p* = 0.54)
MYL6	*Myl6*	Myosin light polypeptide 6	**↑** *p* = 0.0052	**↓** *p* = 0.01144	ns (*p* = 0.75)
OPLA	*Oplah*	5-oxoprolinase	**↑** *p* = 0.04606	**↓** *p* = 0.03249	ns (*p* = 0.52)
PACN2	*Pacsin2*	Protein kinase C and casein substrate in neurons 2 protein	**↑** *p* = 0.03552	**↓** *p* = 0.02471	ns (*p* = 0.99)
PDCD4	*Pdcd4*	Programmed cell death protein 4	**↑** *p* = 0.0012	**↓** *p* = 0.02955	ns (*p* = 0.16)
PDIA1	*P4hb*	Protein disulfide isomerase	**↑** *p* = 0.03143	**↓** *p* = 0.03366	ns (*p* = 0.47)
PDLI5	*Pdlim5*	PDZ and LIM domain protein 5	**↑** *p* = 0.03056	**↓** *p* = 0.03767	ns (*p* = 0.64)
PHLB1	*Phldb1*	Pleckstrin homolog-like domain family B member 1	**↑** *p* = 0.03397	**↓** *p* = 0.01971	ns (*p* = 0.35)
PSB10	*Psmb10*	Proteasome subunit beta type-10	**↑** *p* = 0.00099	**↓** *p* = 0.00199	ns (*p* = 0.97)
PURB	*Purb*	Transcriptional activator protein Pur-beta	**↑** *p* = 0.00037	**↓** *p* = 0.01449	ns (*p* = 0.86)
RHOA	*Rhoa*	Transforming protein RhoA	**↑** *p* = 0.01258	**↓** *p* = 0.00027	ns (*p* = 0.48)
RL13	*Rpl13*	60S ribosomal protein L13	**↑** *p* = 0.00388	**↓** *p* = 0.00536	ns (*p* = 0.59)
RL18	*Rpl18*	60S ribosomal protein L18	**↑** *p* = 0.00386	**↓** *p* = 0.0196	ns (*p* = 0.45)
RL3	*Rpl3*	60S ribosomal protein L3	**↑** *p* = 0.00187	**↓** *p* = 0.00629	ns (*p* = 0.38)
RL35A	*Rpl35a*	60S ribosomal protein L35a	**↑** *p* = 0.03126	**↓** *p* = 0.01536	ns (*p* = 0.31)
ROA2	*Hnrnpa2b1*	Heterogeneous nuclear ribonucleoprotein s A2/B1	**↑** *p* = 0.00755	**↓** *p* = 0.0135	ns (*p* = 0.39)
SPRC	*Sparc*	SPARC	**↑** *p* = 0.044	**↓** *p* = 0.02796	ns (*p* = 0.36)
STIM1	*Stim1*	Stromal interaction molecule 1	**↑** *p* = 0.02664	**↓** *p* = 0.03078	ns (*p* = 0.61)
TAGL2	*Tagln2*	Transgelin-2	**↑** *p* = 0.00181	**↓** *p* = 0.00397	ns (*p* = 0.08)
USO1	*Uso1*	General vesicular transport factor p115	**↑** *p* = 0.01675	**↓** *p* = 0.01326	ns (*p* = 0.88)
WDR1	*Wdr1*	WD repeat-containing protein 1	**↑** *p* = 0.01835	**↓** *p* = 0.01242	ns (*p* = 0.93)
WIPF1	*Wipf1*	WAS/WASL-interacting protein family member 1	**↑** *p* = 0.00052	**↓** *p* = 0.02194	ns (*p* = 0.56)
ARF3	*Arf3*	ADP-ribosylation factor 3	**↓** *p* = 0.03262	**↑** *p* = 0.04184	ns (*p* = 0.23)
CAH1	*Ca1*	Carbonic anhydrase 1	**↓** *p* = 2.30 × 10^−5^	**↑** *p* = 2.53 × 10^−5^	ns (*p* = 0.47)
DCUP	*Urod*	Uroporphyrinogen decarboxylase	**↓** *p* = 0.03293	**↑** *p* = 0.04001	ns (*p* = 0.92)
HEM2	*Alad*	Delta-aminolevulinic acid dehydratase	**↓** *p* = 0.0004	**↑** *p* = 0.03423	ns (*p* = 0.14)
NP1L1	*Nap1l1*	Nucleosome assembly protein 1-like 1	**↓** *p* = 0.00498	**↑** *p* = 0.01454	ns (*p* = 0.53)
BIEA	*Blvra*	Biliverdin reductase A	**↑** *p* = 3.00 10^−5^	**↓** *p* = 0.00726	**↑** *p* = 0.00148
CAN2	*Capn2*	Calpain-2 catalytic subunit	**↑** *p* = 1.09 × 10^−5^	**↓** *p* = 0.02941	*p* = 0.01691
CERU	*Cp*	Ceruloplasmin	**↑** *p* = 2.72 × 10^−5^	**↓** *p* = 0.00447	**↑** *p* = 0.01102
CO3	*C3*	Complement C3	**↑** *p* = 5.32 × 10^−9^	**↓** *p* = 0.00065	**↑** *p* = 1.46 × 10^−5^
CRP	*Crp*	C-reactive protein	**↑** *p* = 1.68 × 10^−7^	**↓** *p* = 0.0023	**↑** *p* = 2.50 × 10^−5^
CSK	*Csk*	Tyrosine-protein kinase CSK	**↑** *p* = 6.65 × 10^−5^	**↓** *p* = 0.00093	**↑** *p* = 0.0141
FIBA	*Fga*	Fibrinogen alpha chain	**↑** *p* = 0.00196	**↓** *p* = 0.00607	**↑** *p* = 0.03133
FIBB	*Fgb*	Fibrinogen beta chain	**↑** *p* = 0.00582	**↓** *p* = 0.02273	**↑** *p* = 7.65 × 10^−5^
FINC	*Fn1*	Fibronectin	**↑** *p* = 2.36 × 10^−5^	**↓** *p* = 0.00042	**↑** *p* = 0.0063
HEMO	*Hpx*	Hemopexin	**↑** *p* = 0.00019	**↓** *p* = 0.0155	**↑** *p* = 0.00854
HSPB1	*Hspb1*	Heat shock protein beta-1	**↑** *p* = 9.81 × 10^−5^	**↓** *p* = 0.02131	*p* = 0.0003
S10A4	*S100a4*	Protein S100-A4	**↑** *p* = 0.00015	**↓** *p* = 0.00305	**↑** *p* = 0.00534
ATOX1	*Atox1*	Copper transport protein ATOX1	**↓** *p* = 0.00085	**↑** *p* = 0.03533	**↓** *p* = 0.02508
B3AT	*Slc4a1*	Band 3 anion transport protein	**↓** *p* = 0.04023	**↑** *p* = 0.00039	**↑** *p* = 0.00089
GYS2	*Gys2*	Glycogen [starch] synthase, liver	**↑** *p* = 0.00021	**↑** *p* = 0.02082	**↑** *p* = 0.0006
PRDX2	*Prdx2*	Peroxiredoxin-2	**↑** *p* = 0.0008	**↑** *p* = 0.02675	**↑** *p* = 0.00086
ARP3	*Actr3*	Actin-related protein 3	**↓** *p* = 0.0345	**↓** *p* = 0.00646	**↓** *p* = 0.00054
CH10	*Hspe1*	10 kDa heat shock protein, mitochondrial	**↓** *p* = 0.0041	**↓** *p* = 0.04784	**↓** *p* = 0.00055
PDIA3	*Pdia3*	Protein disulfide isomerase A3	**↓** *p* = 0.00265	**↓** *p* = 0.00268	**↓** *p* = 4.86 × 10^−5^
PPIA	*Ppia*	Peptidyl-prolyl cis-trans isomerase A	**↓** *p* = 0.02202	**↓** *p* = 0.01276	**↓** *p* = 0.00125
RL13A	*Rpl13a*	60S ribosomal protein L13a	**↓** *p* = 0.02799	**↓** *p* = 0.01762	**↓** *p* = 0.00137
RS16	*Rps16*	40S ribosomal protein S16	**↓** *p* = 0.00333	**↓** *p* = 0.04225	**↓** *p* = 0.0003
RS23	*Rps23*	40S ribosomal protein S23	**↓** *p* = 0.00851	**↓** *p* = 0.01798	**↓** *p* = 6.27 × 10^−5^

**Table 2 ijms-22-08566-t002:** Proteins of the spleen proteome (listed in Table 1) reported in the literature as being associated with curcumin’s therapeutic effects.

	Authors (Year)	Biological Model/Topic
AHR (*Ahr*)	Ciolino et al. [12]	MCF-7 Human breast carcinoma cells
	Rinaldi et al. [13]	Human oral squamous cell carcinoma cells
	Nishiumi et al. [14]	Mouse hepatoma Hepa-1c1c7
	Choi et al. [15]	Hep3B, MCF-7, HEK 293 human cells…
	Garg et al. [16]	Mice (in vivo)
	Cifti et al. [17]	Rats (in vivo)
	Singh et al. [18]	Drosophila larvae
	Mohammadi-Bardbori et al. [19]	Human hepatoma
	Nakai et al. [20]	Mouse hepatoma
ANXA5 (*Anxa5*)	Kam et al. [21]	EAhy926 human endothelial cells
CO9 (*C9*)	Jacob et al. [22]	Mice (in vivo)
GRP78 (*Hspa5*)	Ravindranathan et al. [23]	Colorectal cancer cells
LEG1 (*Lgals1*)	Rabinovitch et al. [24]	Normal rat T cells (from spleen)
	Brandt et al. [25]	Human leukemic T cells
MVP (*Mvp*)	Thiyagarajan et al. [26]	Y79 human retinoblastoma cells
PDCD4 (*Pdcd4*)	Mudduluru et al. [27]	Rko and HCT116 human colorectal cancer cells
	Yang et al. [28]	DU145 human prostate cancer cells, B16 murine melanoma cells
	Chen et al. [29]	Review: microRNAs regulation
	Lelli et al. [30]	Review: melanoma
	Tan et al. [31]	C6 rat experimental glioblastoma
	Shakeri et al. [32]	Review: microRNA-21
PDIA1 (*P4hb*)	Ouyang et al. [33]	Mouse intestinal mucosa (in vivo)
PSB10 (*Psmb10*)	Wang et al. [34]	MCF-7 Human breast carcinoma cells
RHOA (*Rhoa*)	Zhang et al. [35]	Lu1205 and A375 human melanoma cells
	Qin et al. [36]	HSC-T6 rat cells, rat liver fibrosis (in vivo)
	Wang et al. [37]	HMrSV5 human peritoneal mesothelial cells
	Gallardo et al. [38]	MCF-10F and MDA-MB-231 human breast cancer cells
SPRC (*Sparc*)	Kilian et al. [39]	PC-3 human prostate cancer cells + xenografts in CD-1 mice (in vivo)
	Wang et al. [34]	MCF-7 Human breast carcinoma cells
STIM1 (*Stim1*)	Shin et al. [40]	HEK293 human embryonic kidney cells
TAGL2 (*Tagln2*)	Ma et al. [41]	CAL 27 human tongue cancer cells

## Data Availability

Not applicable.

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
