# Peer review of "Lymphoid Organ Proteomes Identify Therapeutic Efficacy Biomarkers Following the Intracavitary Administration of Curcumin in a Highly Invasive Rat Model of Peritoneal Mesothelioma"

_ijms, 2021, doi:10.3390/ijms22168566_

Round 1
Reviewer 1 Report
Comments to authors:
The manuscript mainly focused on curcumin-induced proteomics changes in a rat model system. Is there any limitation with the lines of title? If not, the authors need to replace the words bi-omarkes and curcu-min with biomarkers and curcumin, respectively. This applies throughout the abstract and manuscript. Although curcumin effects have been demonstrated by other cancer studies, the experimental design of this study looks good to me. The authors shown the localization of eighty target proteins in way easy to understand the curcumin-induced alterations. The identification of twelve targets of curcumin that have already been published by other studies support the results of this study. However, I did not see any orthogonal experiments such as western blot or ELISA to confirm the expression levels identified by proteomics analysis. Are there any reasons to conclude the results based on the proteomics experiments only? It is required to provide sonication details in the experimental section of proteomic analysis (section 4.3). Overall, the authors well described their experiments and results. However, proofreading with a native English speaker is required to improve the text of this manuscript.
Reviewer 2 Report
This study concentrated on the identification of proteomic changes as curcumin treatment in the malignant mesothelioma rat model. They compared histological sections after proteome analysis and found some biomarkers and crucial targets in anti-cancer effects. Moreover, they discovered the changes in circulating markers, C-reactive protein, and complement C3... and so on.
Briefly, this study is an exciting investigation but still has some issues that need to be clarified.
1) The authors should provide images of the tumor mass (with and without treatment) in mice and a detailed description of the tumor collection process in the method section.
2) The authors need to provide more information in Table 1, such as p-value and FDR adjusted p-value.
3) The authors should perform GO or enrichment analysis and functional STRING network in Figure 2.
4) The validation of abundance change proteins(Ahr, Pdcd4, and Rhoa) in the spleen should be manipulated by western blot.
5) The X-axis label in Figure 4, Figure 5, Figure 6, and Figure 7 is too small. The authors should enlarge them.
6) The validation of abundance change proteins PNPH in the lymph node should be manipulated by western blot.
Round 2
Reviewer 2 Report
I agree to publish this paper in IJMS.